# Plant-Growth Endophytic Bacteria Improve Nutrient Use Efficiency and Modulate Foliar N-Metabolites in Sugarcane Seedling

**DOI:** 10.3390/microorganisms9030479

**Published:** 2021-02-25

**Authors:** Matheus Aparecido Pereira Cipriano, Raquel de Paula Freitas-Iório, Maurício Rocha Dimitrov, Sara Adrián López de Andrade, Eiko Eurya Kuramae, Adriana Parada Dias da Silveira

**Affiliations:** 1Centro de Solos e Recursos Agroambientais, Instituto Agronômico, IAC, 13020-902 Campinas, São Paulo, Brazil; raquel.p.f@hotmail.com; 2Microbial Ecology Department, Netherlands Institute of Ecology, NIOO-KNAW, 6708 PB Wageningen, The Netherlands; mau.dimitrov@gmail.com (M.R.D.); e.kuramae@nioo.knaw.nl (E.E.K.); 3Departamento de Biologia Vegetal, Universidade Estadual de Campinas, UNICAMP, 13083-970 Campinas, São Paulo, Brazil; sardrian@unicamp.br; 4Ecology and Biodiversity, Institute of Environmental Biology, Utrecht University, 3584 CH Utrecht, The Netherlands

**Keywords:** *Saccharum* sp., plant-growth promoting bacteria, nitrogen, endophytes, amino acids, enzymatic activity, plant nutrition

## Abstract

Beneficial plant–microbe interactions lead to physiological and biochemical changes that may result in plant-growth promotion. This study evaluated the effect of the interaction between sugarcane and endophytic bacterial strains on plant physiological and biochemical responses under two levels of nitrogen (N) fertilization. Six strains of endophytic bacteria, previously selected as plant growth-promoting bacteria (PGPB), were used to inoculate sugarcane mini stalks, with and without N fertilization. After 45 days, biomass production; shoot nutrient concentrations; foliar polyamine and free amino acid profiles; activities of nitrate reductase and glutamine synthase; and the relative transcript levels of the *GS1*, *GS2*, and *SHR5* genes in sugarcane leaves were determined. All six endophytic strains promoted sugarcane growth, increasing shoot and root biomass, plant nutritional status, and the use efficiency of most nutrients. The inoculation-induced changes at the biochemical level altered the foliar free amino acid and polyamine profiles, mainly regarding the relative concentrations of citrulline, putrescine, glycine, alanine, glutamate, glutamine, proline, and aspartate. The transcription of *GS1*, *GS2*, and *SHR5* was higher in the N fertilized seedlings, and almost not altered by endophytic bacterial strains. The endophytic strains promoted sugarcane seedlings growth mainly by improving nutrient efficiency. This improvement could not be explained by their ability to induce the production of amino acid and polyamine composts, or *GS1*, *GS2*, and *SHR5*, showing that complex interactions may be associated with enhancement of the sugarcane seedlings’ performance by endophytic bacteria. The strains demonstrated biotechnological potential for sugarcane seedling production.

## 1. Introduction

Throughout their evolution, plants have developed a complex set of mechanisms for environmental adaptation. One such mechanism is the association with beneficial microorganisms, such as endophytic and rhizosphere bacteria known as plant-growth-promoting bacteria (PGPB). PGPB have been explored as pathogen antagonists and biostimulants of plant growth, suggesting an ecofriendly alternative to pesticides and chemical fertilizers in a sustainable agriculture [1]. The plant–bacteria interaction, through a complex array of mechanisms, can result in plant growth-promotion due to increased nutrient uptake, nitrogen fixation, or phytohormone production, or indirectly due to the phytopathogen suppression [2]. The nutritional benefits and growth stimulation can be directly linked to the improvement in plant nitrogen (N) status by biological nitrogen fixation (BNF) activity, triggered by diazotrophic bacteria which possess the *nif* gene, or to the production of auxin-related compounds, and also to other mechanisms such as phosphate solubilization and siderophore production—all desirable characteristics of PGPB [3,4]. N uptake from soil can occur either in the nitrate or ammonia forms or be provided by diazotrophic bacteria in the form of ammonium (NH_4_^+^). Thus, N assimilation enzymes, such as nitrate reductase (NR) and glutamine synthetase, have been assessed to evaluate endophytic bacteria influence on N uptake and use efficiency, indicating activation of N metabolism and a higher uptake from soil [5,6,7]. Plant receptors of bacterial signals are known to recognize phytopathogenic bacteria or be involved in the identification of beneficial microorganisms by plants [8,9]. The repression of *SHR5* transcription, a receptor-like kinase located in plant membrane, has been used as a proxy of beneficial association between N_2_-fixing endophytic bacteria and plants [8]. Likewise, the role of amino free acids and polyamines has been shown to alter during the interaction between plants and microorganisms. Both plant-phytopathogenic bacteria and plant-beneficial microorganism interaction may result in significant changes in polyamine metabolism of the host and/or microbe partners, revealing it to be a complex and dynamic process [10]. Despite evidence of polyamines’ role in plant–pathogen interaction, their role in plant-beneficial microbe associations has been related to the course of association establishment [10,11].

PGPB inoculation is recommended for sugarcane to improve sustainable alternatives to the use of synthetic fertilizers, especially with endophytic bacteria strains of the genera *Azospirillum*, *Kosakonia*, *Herbaspirillum*, *Paraburkholderia*, and *Pseudomonas* [10,12,13]. Plant genetic factors may contribute to the increased efficiency of plant–bacteria interaction, causing plant physiological changes that culminate in the modulation of plant growth and development [14,15]. Polyamines, besides their function in developmental processes, such as cell division, elongation, and organogenesis, respond to environmental cues, being involved in plant response to biotic and abiotic stressors [10].

Nowadays, Brazil is the largest worldwide sugarcane producer, and the management of this crop has changed due to the expansion of its cultivation frontiers in the country. However, one of the biggest challenges in sugarcane production is maintaining high productivity and minimizing the harmful environmental effects of low fertilization efficiency and long-term use of the soil, which can favor the selection of harmful microorganisms that can reduce agricultural production. In this context, inoculants based on endophytic bacteria and/or rhizobacteria strains suggest a sustainable alternative to this form of agriculture. Sugarcane seedling production, one of the main stages of this crop production, is conducted in nurseries via micropropagation or mini-stalks using organic substrates, which can favor beneficial bacteria inoculation. It is known that endophytic microorganism communities that inhabit sugarcane tissues are more diverse than previously thought [16]. This high microbial diversity challenges the introduction of new bacterial inoculants due to a high competition for physical space and niches, justifying the importance of the search for different and more efficient bacterial plant-promoting strains able to colonize sugarcane roots. Recent studies recognized the endophytic bacteria *Paraburkholderia caribensis* IAC/BECa-088, *Paraburkholderia tropica* IAC/BECa-135, *Kosakonia radicincitans* IAC/BECa-095, *Pseudomonas fluorescens* IAC/BECa-141, and *Herbaspirillum frisingense* IAC/BECa-152 as suitable inocula for sugarcane cultivation with the potential for alleviating Al stress in sugarcane plantlets [6,13,17]. These bacteria possess desirable characteristics of PGPB, such as antagonistic activities against phytopathogens, siderophore, auxins, and cyanide production. The strains IAC/BECa-135, IAC/BECa-141, and IAC/BECa-152 are negative for *nif*H genes and strains IAC/BECa-088, IAC/BECa-090, and IAC/BECa-095 amplify the *nif*H gene. We hypothesize that (a) endophytic bacterial strains with different characteristics differently alter the plant response regarding N metabolism, resulting in plant growth promotion and adequate nutrient balance; and (b) this plant response is more adequate at low soil N. In order to test this hypothesis we designed an experiment to evaluate if the sugarcane seedlings inoculated with endophytic bacteria could reduce the need for N fertilization. 

Increasing environmental concerns and the search for a more sustainable agriculture have led research to intensify the development of biofertilizers. However, the correct and efficient use of microorganisms as inoculants requires more knowledge about their benefits and impacts. Thus, the objective of this study was to evaluate the effect of bacterial inoculants on the development of sugarcane plants, and the physiological and biochemical aspects of plant–bacteria interaction.

## 2. Material and Methods

### 2.1. Bacterial Strain Inocula Preparation

All six strains obtained from Agronomic Institute (IAC, Campinas, Brazil) Culture Collection were isolated from the root endosphere of healthy sugarcane plants (*Saccharum officinarum* L.). These strains were previously characterized for plant growth traits such as antagonistic activities against phytopathogens, indole-3-acetic acid (IAA) production, PCR amplification of the *nifH* gene, and phosphate solubilization, besides improving sugarcane plant growth in previous researches [6,13,18,19,20], as summarized in Table 1. The bacterial strains were grown in DYGS medium for 72 h at 28 °C. Each inoculum was centrifuged (12,000 rpm, 10 min), suspended in a sterile solution of 0.01 mol·L^−1^ MgSO_4_·7H_2_O and adjusted to a density of 10^8^ colony forming units (CFU)·mL^−1^. The method and density were the same for all bacteria inoculation during the experiment. 

### 2.2. Evaluating the Sugarcane Seedlings Growth under Greenhouse Conditions

The experiment was performed with sugarcane mini-stalks, variety IACSP 95–5000, at the Agronomic Institute (22°54′20″ S, 47°05′34″ W). Sugarcane mini-stalks were cultivated in pots (200 mL), with a sterile commercial substrate (Tropstrato^®^), and maintained under greenhouse conditions for three weeks. During this period, two bacterial inoculations, of 2 mL each, were carried out, on the base of the stems at the planting time and 7 days later. After three weeks (T0), some of the seedlings were harvested and evaluated for plant biomass and the remaining seedlings were reinoculated with the respective strains (10 mL) and after 24 h transferred to pots with 3 L of soil (Red Oxisol). The soil chemical analysis is described in Appendix A; it was fertilized as recommended by van Raij et al. [20] with some modifications, especially for nitrogen (N). Half of the seedlings were not fertilized with N (to evaluate the possible BNF by the diazotrophic bacteria strains) and the other half was fertilized as described: 350 mg of N per pot as NH_4_NO_3_ (one third at planting and two-thirds after 30 days). The other two samples were taken at 30 (T30) and 45 (T45) days after transplanting. All the plants were fertilized with other nutrients as described: 100 mg kg^−1^ of P, as simple superphosphate (half at planting and half after 30 days); 100 mg kg^−1^ of K, as KH_2_PO_4_ (one third at planting, and two-thirds after 30 days); 200 mg kg^−1^ of Mg, as MgSO_4_; 5 mg kg^−1^ of B, asH_3_BO_3_; 10 mg kg^−1^ of Zn, as ZnSO_4_; 5 mg kg^−1^ of Cu, as CuSO_4_; and 1 mg kg^−1^ of Mo, as Na_2_MoO_4_.

The experimental design was completely randomized with six replicates, in a 7 × 2 factorial scheme, with the causes of variation being bacterial treatment (six bacterial strains: IAC/BECa-088, IAC/BECa-090, IAC/BECa-095, IAC/BECa-135, IAC/BECa-141, IAC/BECa-152, and a control without inoculation) and fertilization condition (with and without N fertilization).

### 2.3. Evaluation of Plant Growth-Promotion and Plant Nutritional Status

At harvest, at T0, T30, and T45, shoots and roots were separated and evaluated the shoot and root dry weight was obtained, after air-drying at 60 °C to constant weight. Plant shoot macro and micronutrients concentration were determined according to Bataglia et al. [21]. The nutrient accumulation (NA) and nutrient-use efficiency index (UEI) were estimated according to Siddiqi and Glass [22], following calculations: nutrient accumulation (NA) = nutrient concentration × plant biomass and UEI = (plant biomass)^2^/NA.

### 2.4. Enzyme Extraction and Enzyme Activities Determination

At T45, leaves +2, according to the system of Casagrande [23], were collected and immediately frozen in liquid nitrogen and stored at −80 °C until enzyme activity determination. The activity of the enzymes nitrate reductase (NR), nitrite reductase (NiR), and glutamine synthetase (GS) was performed, with some modifications, as described by Silveira et al. [24].

Leaf samples were finely macerated with liquid nitrogen and PVPP and 5 mL of the extracting buffer (100 mM Tris-HCl pH 7.5, 10 µM FAD, 20 mM EDTA, 5 mM DTT, 0.5% BSA, 0.1 mM PMSF,1 mM benzidine) was added. The homogenates were filtered through two layers of gauze and centrifuged at 10,000 rpm at 4 °C for 20 min. The supernatant extract was collected for enzyme activity determination. 

### 2.5. Activity of Nitrate Reductase Enzyme (NR)

Enzyme extract (200 µL) was added to 500 µL of reaction buffer (100 mM Tris-HCl pH 7.5, 10 mM EDTA, 5 mM KNO_3_, 5 mM DTT, 10 µM FAD) and 15 µL of 1 mM NADH in each microtube and incubated in a water bath at 30 °C for 30 min. The reaction was stopped by placing the microtube in a water bath at 100 °C for 10 min and then 750 µL of sulfanilamide (1% sulfanilamide and N-1-naphtyl dissolved in 2.4 N HCl) was added. To eliminate the turbidity, the reaction mixture was centrifuged, the supernatant collected and read spectrophotometrically at 540 nm.

### 2.6. Activity of Nitrite Reductase Enzyme (NiR)

300 µL of enzyme extract was added to 1.5 mL of the buffered reaction mixture: 100 mM Tris-HCl pH 7.5, 15 mM NaNO_2_, 5 mM methylviolagen). The reaction was started by addition of 200 µL of a freshly prepared solution of 86 mM Na dithionite dissolved in a 190 mM NaHCO_3_ solution. After incubation at 30 °C for 15 min, the reaction was stopped by very vigorous shaking. The amount of nitrite was determined as reported for NR activity assay. 

### 2.7. Activity of Glutamine Synthetase Enzyme (GS)

100 µL enzyme extract were added to 300 µL of reaction buffer (50 mM Tris-HCl pH 7, 0.5 mM MgSO_4_, hydroxylamine100 mM NaOH, 50 mM glutamate), and 100 µL of ATP 100 mM. The reaction was incubated at 30 °C for 30 min and 500 µL of ferric solution (FeCl_3_ 0.37 M, TCA 0.2 M dissolved in 0.67 M HCl) was added to stop the reaction. To eliminate the turbidity the reaction mixture was centrifuged and the supernatant spectrophotometrically read at 535 nm.

### 2.8. RNA Extraction and Quantitative Real-Time PCR (RT-qPCR) Analysis

At T45 harvest, +1 leaves, defined as those with the most recently exposed collar, were collected and immediately frozen in liquid nitrogen and stored at −80 °C till RNA extraction. Based on the best results of growth and nutrient uptake promotion only four bacterial inoculation treatments were selected for the analysis of three gene expression level: control, IAC/BECa-088, IAC/BECa-095, and IAC/BECa-141, with and without N fertilization, comprising 8 treatments.

Total RNA was extracted from approximately 100 mg of macerated leaves using the *RNeasy Plant Mini Kit* (Qiagen, Hilden, Germany), in compliance with the manufacturer’s recommendations, followed by treatment with RNase*-Free DNase Set* to remove any contaminating genomic DNA. cDNA synthesis was performed using the *Easy Script First-Strand cDNA Synthesis SuperMix* (TransGen Biotech, Beijing, China) for RT-PCR according to the manufacturer’s recommendations.

For the analysis of gene expression, three target genes were analyzed, *GS1*, *GS2*, codifying for glutamine synthetase 1 and 2 isoforms, found in the cytoplasm and chloroplasts, respectively, and *SHR5*, relative to a receptor-like kinase involved in plant-endophytic recognition. Three constitutive genes were tested, *GAPDH*, *UBQ*, and *TUB,* and then *GAPDH* was selected as the most stable among different treatments [25]. For *GS1*e *GS2* analysis it was used the sequences already deposited in databases [26], and for *SHR5* program *Primer3* (www.bioinfo.ut.ee/primer3-0.4.0, accessed on 11 February 2021) to design primers, followed by checking formation of structures in the program *OligoAnalyzer 3.1* [27] and hybridization check in the program BLAST in NCBI [28]. The set primers used were GS1-F (CATCGAAGCTGTTGAGGACA), GS1-R (ACGCGGCATCTATATTGACC); GS2-F (ATCCTCCATCTTCACGCATC), GS2-R (ATCCTCCATCTTCACGCATC); SHR5-F (TATTCCTCCTTTGCCGTAG), SHR5-R (CACCCCATCTTGTTTGACC); and GAPDH-F (CACGGCCACTGGAAGCA), GAPDH-R (TCCTCAGGGTTCCTGATGCC). The PCR procedure was 95 °C for 3 min; 40 cycles of 95 °C for 10 s; and 60 °C for 30s with a final extension at 72 °C for 4 min.

The first-strand cDNAs were diluted 50 times with nuclease-free water and used in RT-qPCR reactions. The reactions of RT-qPCR were prepared with *QuantiFastTM SYBR Green PCR Ki*t (Qiagen, Hilden, Germany) and analyzed in a thermocycler *StepOnePlus Real-Time PCR System* (Applied Biosystems, Waltham, MA, USA). Each reaction contained 1.6 μL of nuclease-free water, 0.2 μL of each primer (1.25 pmol mL^−1^), 5.0 μL of the reagent, and 3 μL of diluted cDNA. The analysis of transcripts quantification was performed by quantification technique on the method of ΔΔ*C*t [29].

### 2.9. Profile of Amino Acids and Polyamines

The determination of the profile of free amino acids and polyamines was performed by ultra-performance liquid chromatography mass spectrometry (UPLC-MS). Samples of 1+ leaves were collected at T45, macerated in liquid nitrogen and lyophilized. Extraction was performed from 100 mg of lyophilized material, homogenized with 1 mL of methanol water (80:20, *v*:*v*) in an ultrasonic bath at 30 °C for 15 min, centrifuged at 10,000 rpm for 5 min and the supernatant collected for analysis. The extracted samples were analyzed directly on an *Acquity UPLC-MS* (QTOF, *Micromass-Waters*, Manchester, UK). The chromatographic separation was carried in a Waters *Acquity C_18_ BEH analytical column* (150 mm × 2.1 mm i.d, 1.7 µm) at 30 °C. Methanol (A) and 0.1 % formic acid in water (B) were used as mobile phase at a flow of 0.2 µL·min^−1^. The initial condition gradient was 1% of A and 99% of B up to 2.5 min., followed for 50% of A and 50% of B. This condition was maintained until 5 min., returning to the initial condition and stabilizing at 8 min. The analysis in the mass spectrometer were carried out using electrospray ionization in negative mode under the following conditions: capillary 3.0 kV and cone 30 V, temperature in the ionization source of 150 °C and desolvation temperature 350 °C, and CID of 15 ev. Output data were obtained in the range between 50 at 300 *m/z*. For quantification of amino acids and phenolic compounds, calibration curves were produced by the injection of known concentration of the standards [30].

### 2.10. Statistical Analysis

All analytical assays were performed in triplicate and data were submitted to analysis of variance (Two way-ANOVA), followed by the comparison of means by the Scott-Knott test at 5%, using the software SISVAR [31].

## 3. Results

### 3.1. Plant Growth-Promotion and Nutritional Status of Plant

All six strains inoculated on sugarcane seedlings improved sugarcane growth and had a significant effect on bacterial inoculation and N fertilization (Figure 1), but the interaction between the factors was not significant (data not shown). At T0, the seedlings inoculated with endophytes strains produced significantly higher shoot biomass than plants without inoculation and there were no significant differences in root biomass production among treatments (Table 2). After 30 days of transplanting (T30), the effect of the bacteria on growth was significant in shoot, while the inoculation of IAC/BECa-088 also promoted seedling root growth. At T45, after 45 days of transplanting, growth promotion by bacterial strains was evident in shoot and root (Table 2/Figure 1A). N fertilization (Figure 1B) increased root biomass production at both evaluation times and shoot and total biomass production at T45 and there were no significant differences in the number of formed tillers due to N fertilization.

The inoculation of bacterial strains led to significant changes in the concentration, accumulation, and use efficiency indexes of most plant nutrients (Table 3 and Table 4). Nutrient uptake and accumulation were not significantly influenced by the application of N fertilizer and the interaction between bacterial and N fertilization treatments was also not significant (data not show). For these reasons, just the results regarding to bacterial inoculation are shown (Table 3 and Table 4). At T30, the inoculation of bacterial strains did not influence Mg, S, and Mn nutrient concentration in plant shoots (Table 3), and at T45 there were no significant differences in shoot Mg nutrient concentrations of plants inoculated or not with the different bacterial strains (Table 4). Regarding the shoot nutrient accumulation, it was observed that, at T30 (Table 3), all bacterial strains, except IAC/BECa-152 (*H. frisingense*), showed a significant increase in cumulative amounts of all nutrients, except for B, compared to the control treatment. The bacterial strain IAC/BECa-152 promoted larger accumulations of N, Mg, Fe, and Mn. At T45, the inoculation of all bacterial strains promoted an increase in nutrient accumulations, except for Fe, for strain IAC/BECa-141, and P, which only increased in plants inoculated with IAB/BECa-090 strain (Table 4).

Concerning the nutrient-use efficiency index (UEI), at T30 (Table 3), the bacterial inoculation, except IAC/BECa-095, promoted significant increases in UEI, for all analyzed nutrients when compared to control, except for B. The UEI for Fe in plants treated with strain IAC/BECa-095 did not differ significantly from that observed in control plants. At T45 (Table 4), the significant increase in NUE was also observed for all the bacterial strains, except in plants treated with IAC/BECa-152, that exhibit only higher UEI for K, Ca, and Zn. Only plants treated with IAC/BECa-141 showed increase in UEI for B.

### 3.2. Enzymatic Analysis—Nitrogen Metabolism Enzymes (NR, NiR and GS)

The activities of NR, NiR and GS in leaves were not significantly influenced by bacterial inoculations (Appendix A), N fertilization, or the interaction between them (data not shown).

### 3.3. Relative Expression of the Genes GS1, GS2, and SHR5

N fertilization significantly decreased the expression of the gene *GS1* (cytosolic isoform) in leaves of plants inoculated with all bacterial strains (Figure 2). The same effect was observed for *GS2* gene (plastidic isoform), except in plants inoculated with the strain IAC/BECa-095. In bacterial treatments, it was observed that, without N fertilizer addition, the leaves of the plants treated with IAC/BECa-095and IAC/BECa-141 showed lower relative abundance of *GS2* transcripts, whereas expression of the *GS1* gene did not change among plants with different bacterial strains inoculation. The relative expression of *GS1* was higher than that observed for *GS2* in all treatments.

The addition of N to soil decreased the relative expression of the gene *SHR5* (receptor-like kinase) in leaves of control plants and those inoculated with IAC/BECa-141 strain (Figure 2). In plants without N fertilization and treated with IAC/BECa-095 and IAC/BECa-141, the expression of this gene was lower when compared to control plants. With N addition, there was no significant difference among them.

### 3.4. Free Amino Acid and Polyamines Profiles

Concerning the foliar free-amino acid and polyamine profiles, a significant effect of bacterial strains inoculation was observed (Figure 3; Appendix A). The inoculation of bacterial strains did not significantly alter the levels of Lys, Gaba, Thr, Val, and Phe, but caused a significant decrease in the levels of the polyamine putrescine (Appendix A). Compared with control treatment and depending on the bacterial strain, a significant increase was observed in the contents of Gly (IAC/BECa-090, IAC/BECa-095, IAC/BECa-135 and IAC/BECa-152) and Ala (IAC/BECa-088 and IAC/BECa-095), and decreases in the levels of free Glu (IAC/BECa-141 and IAC/BECa-152), Asp(IAC/BECa-135, IAC/BECa-141 and IAC/BECa-152), Gln(IAC/BECa-088, IAC/BECa-095 and IAC/BECa-141), Pro (IAC/BECa-090, IAC/BECa-135, IAC/BECa-141 and IAC/BECa-152), and citrulline (IAC/BECa-090, IAC/BECa-95, IAC/BECa-135, IAC/BECa-141 and IAC/BECa-152).

## 4. Discussion

Positive interaction between plant and microorganisms at the rhizosphere or endosphere compartments has been recognized to occur in several environmental conditions improving plant performance [13,32]. This study verified the beneficial effect of the inoculation of some strains of endophytic bacteria on the growth and nutritional status of sugarcane seedlings. 

The growth-promoting effect on sugarcane seedlings was observed as early as 15 days after the inoculation of the bacterial strains at seedling formation stage, and maintained through the cultivation period until 45 days after transplanting to the soil. Plant growth-promotion has already been observed as a positive consequence of interactions between endophytic and diazotrophic bacteria and plants of the Poaceae family, such as sugarcane [7,33], maize and wheat [5,32], or rice [34,35]. This effect has been observed in several experiments, including some under field conditions [36,37,38]. Gírio et al. [39] evaluated the effect of the inoculation of a mix of five PGPB species: *Gluconacetobacter diazotrophicus, Azospirillum amazonense, Burkholderia tropica, Herbaspirillum seropedicae*, and *Herbaspirillum rubrisubalbicans*, in sugarcane seedlings, with and without N fertilization, obtaining similar results to those in our experiment, as the inoculation promoted gains in plant growth, regardless of N application.

Besides the positive influence on sugarcane growth, nutrient uptake and accumulation were significantly increased in inoculated seedlings, showing that the studied bacterial strains might possess a set of mechanisms by which they improve plant nutrient uptake capabilities, as observed previously by Prieto et al. [40]. The association with PGPB can influence plant nutrient acquisition, either by increasing nutrient availability in the rhizosphere or influencing the physiological mechanisms underlying nutritional processes [41]. Such mechanisms may include changes in root system architecture, altering shoot to root biomass ratio, or changes in root physiology, increasing proton efflux by modulating H^+^-ATPases activity or by promoting organic anions exudation and enhancing nutrient solubility, or yet by the indirect effect of IAA produced by PGPB [41].

This increase in nutrient concentration and use efficiency has been reported in sugarcane inoculated with *G. diazotrophicus* strains [42], and tomato inoculated with *Brevundimonas* spp. and *Micrococcus* spp. [43], but not to the extent observed in this study, in which practically all nutrients showed an increase in nutrient accumulation, especially for N, P. K, Ca, Mg, and S UEI (Figure 4), when plants were inoculated with bacteria strains. The increase in nutrient UEI reflects that inoculated seedlings were more efficient than non-inoculated plants in producing biomass per unit of nutrient taken up [44]. This efficiency relies on the ability to absorb nutrients from the soil and also on plant utilization and remobilization processes [40]. In this study all bacterial strains had a clear positive effect on root growth, when compared to the non-inoculated seedlings. Root growth promotion and changes in root morphology and architecture by endophytic bacteria have been linked to increased nutrient uptake capabilities by Prieto et al. [40].

Both growth promotion and improvement of plant nutritional status may be related to efficient plant–bacteria interaction [45]. Many of these microorganisms are described as producers of plant growth regulators, such as auxin, cytokinins, and gibberellins [46,47,48]. Some PGPB can increase the solubility of phosphates and zinc sources, as well as enhancing siderophore production, known to assist Fe acquisition by roots [48,49]. The production of growth regulators can alter growth dynamics and root nutrient uptake, modifying root morphology and changing nutrient acquisition. Root growth benefits the absorption of nutrients and water by increasing the contact surface area, however, these bacteria can also stimulate the uptake systems and ion transport through mechanisms not yet clearly defined [50,51,52]. 

The endophytic bacteria strains used in this study are facultative endophytic, and probably they also colonize the rhizosphere of sugarcane. Thus, it can be speculated that, during the colonization process, a population remained in the rhizosphere which may have contributed to further nutrient solubilization, making them more available for plant uptake as previously observed by Kuklinsky-Sobral et al. and Postma et al. [53,54].

Growth promotion by endophytic bacteria can be also related to their BNF capability and its influence on plant N metabolism [55]. In this study, three of the PGPB strains, IAC/BECa-088, IAC/BECa-090, and IAC/BECa-095, possess the *nifH* gene, encoding the Fe-protein component of the nitrogenase enzyme [56]. However, while there was no gain in plant N contents or the activities of N metabolism-related enzymes (NR, NiR, and GS), N use efficiency was significantly higher in seedlings inoculated with bacteria (Figure 4). Interestingly, this higher N use efficiency was also observed for bacteria lacking the *nifH* gene. The contribution to plant development by non-N_2_-fixing endophytic bacteria has been highlighted by other authors [57,58], suggesting that other growth-promoting mechanisms are induced by PGPB that lead to a better plant nutritional status [40,48]. In general, N fertilizer addition did not cause a significant influence in most of the analyzed variables, probably, due to the adequate fertility of the soil used in this experiment and to the short period between fertilization and sampling. 

Enzymatic assays were performed by an in vitro method which, unlike in vivo, breaks cells, and provides artificial conditions for enzyme activity, with optimum concentrations of substrate, reducing power (NADH and FAD) and protective substances, thus showing potential activity that may not be correlated with in planta activities [24]. The activity of the GS depends on the environmental conditions and it is regulated at the transcriptional level [59,60]. GS is a key enzyme in N assimilation, metabolism, and remobilization and occurs in two forms: the cytosolic isoform, *GS1*, related to the synthesis and transport of glutamine by primary N assimilation and to the recycling of NH_4_^+^ released by processes such as protein or amino acid catabolism, during N remobilization; and the chloroplastic isoform, *GS2*, involved in both primary assimilation and in the re-assimilation of NH_4_^+^ generated by photorespiration [61]. It was observed that the relative expression of *GS1* was higher than that observed for *GS2* (Figure 1). The high abundance of *GS1* transcripts in sugarcane leaves suggests a key role of this isoform in N metabolism of C4 plants [26]. Both genes decreased their expression levels under N fertilizer application.

A decrease in GS isoforms in *Arabidopsis* plants grown at high N concentration was observed, and their activity increased at plant senescence, facilitating the plant’s N remobilization [62]. In sugarcane, Nogueira et al. [26] observed that increased N availability resulted in increased expression of these genes in some varieties, but the authors point out differences in plant genotype performance. Different results could be related to genotypic variation, the tissue analyzed, and N availability. In this study, gene expression analysis was performed in the leaf +1 [23], during the stage of transition to maturity, as it began to remobilize N to younger leaves at the apical part, which may explain the higher transcription levels of *GS1* than in relation to *GS2*, and also the higher expression levels of this enzyme under lower N conditions, where remobilization of N reserves could be important [63,64].

In plants growing in the absence of N fertilizer, the inoculation of the bacterial strains (IAC/BECa-095 and IAC/BECa-141) caused lower abundance of *GS2* transcripts than plants that did not receive inoculation. The plastidial *GS2* is related to the assimilation of NH_4_^+^ produced by NR/NiR activity or/and during photorespiration, and has been shown to be more active under stressing conditions [65,66]. The lower adundance of the *GS2* transcripts and higher Gly contents may be related to lower Gly decarboxylation through photorespiratory pathways in inoculated seedlings [67]. As a C4 plant, sugarcane is not expected to show significant oxygenase activity of the rubisco, minimizing the role of *GS2* in photorespiratory NH_4_^+^ recycling. 

The expression gene *SHR5*, encoding a receptor-like kinase, has been intimately linked to the recognition of beneficial microorganisms by plants, and repression of *SHR5* expression due to plant–pathogen interaction or in response to abiotic stresses has not been observed, indicating that its down-regulation is directly linked to the established success of beneficial interactions [8,14]. Its specific function has not yet been determined, but members of this family of receptors are associated with important roles in the development symbiosis and plant defense mechanisms [68,69,70]. In the present study, a decrease in the expression of the *SHR5* gene was observed in plants inoculated with some of the bacterial strains (IAC/BECa-095 and IAC/BECa-141) in the absence of N fertilization, suggesting an efficient and beneficial association among these strains and sugarcane. Vinagre et al. [8] observed a decrease in the expression of this gene in the presence of beneficial endophytes in sugarcane, suggesting that the expression levels of this gene are inversely related to the efficiency of plant–bacteria association.

Among various alterations in the plant physiology brought by plant–microbe interactions, changes in amino acid metabolism and amino acid contents have been prevailing observations [71]. The polyamine putrescine is closely related to the metabolism of amino acids, organic acids, and also with hormonal cross-talks [67]. The foliar free amino acid and polyamines profiles were modified by the inoculation of the bacterial strains. Tejera et al. [72] verified a decrease in most free amino acids and polyamines in sugarcane plants associated with the endophytic bacteria *G. diazotrophicus*. Weston et al. [71] reported changes in free amino acids and other metabolites in *Arabidopsis* plants after *P. fluorescens* inoculation. In our study, we also observed lower relative contents of most foliar free amino acids when plants were associated with endophytic strains. The accumulation of some amino acids might be an indication of physiological stress, as accumulation proline, aspartate, and putrescine was observed [73,74,75]. It was also evident that there were significantly higher proportions of Ala in the leaves of plants inoculated with IAC/BECa-88 and IAC/BECa-95, which both have BNF capabilities. Increases in Ala contents have been related to carbon assimilation and metabolism in C4 plants, as one of the major exchange transport metabolites between mesophyll and the bundle sheath [76]. In plants inoculated with the strains IAC/BECa-90, IAC/BECa-95, IAC/BECa-135, and IAC/BECa-142, glycine contents were significantly higher than in control plants, and represented more than 20% of total free amino acids in leaves of plants inoculated with IAC/BECa-90, IAC/BECa-135, and IAC/BECa-152. Glycine is involved in photorespiratory metabolism, and its accumulation may reflect lower oxygenase activity of the rubisco in these plants [67]. The polyamine putrescine has been frequently related to the response of plants to different abiotic and biotic stresses, with increased accumulation conferring normally higher tolerance to the stressor [75]. Here, we found a significantly lower accumulation of putrescine in inoculated plants compared to control plants, suggesting a higher putrescine metabolism towards higher polyamines or other catabolites [77], nonetheless, different physiological roles are attributed to polyamines and they can be involved in signaling during stress, antioxidant response, or nutrient transport [77]. Some evidence also shows the importance of polyamines for plant-beneficial microorganism interaction, such as nodule organogenesis on legume roots triggered by rhizobia bacteria and mycorrhizae formation in plant tissues [10].

## 5. Conclusions

The bacterial strains positively influenced sugarcane seedlings growth, showing benefits as early as fifteen days after inoculation. The strains improved plant nutritional status and seedlings showed higher contents of most of the nutrients and higher nutrient use efficiency, which probably contributed to the higher biomass production in inoculated sugarcane seedlings. Sugarcane–endophyte interaction influenced plant N metabolism, and free amino acid and polyamine contents in leaves. The inoculation of some strains increased foliar alanine and glycine contents, and decreased the contents of proline, aspartate, and putrescine, considered stress-related compounds, suggesting that PGPB-inoculated plants could be under lower stress conditions than non-inoculated plants. Some bacterial strains also altered the expression of *GS2* and *SHR5* genes in leaves of sugarcane. The SHR5 involvement in defense response signaling, and the lower levels of its transcripts in inoculated plants, might be related to the establishment of symbiotic association between sugarcane seedlings and endophytic bacteria. Overall, independently of N availability, and of the diazotrophic capability of the strains, the endophytic bacteria improved nutrient use efficiency and growth and modulated foliar N-metabolites in sugarcane seedlings. 

Finally, the ability of the endophytic strains in promoting sugarcane growth may be attributed to an array of mechanisms that modulate N metabolism and nutrient use efficiency. The results reveal the biotechnological potential of these selected strains for sugarcane seedling production.

## Figures and Tables

**Figure 1 microorganisms-09-00479-f001:**
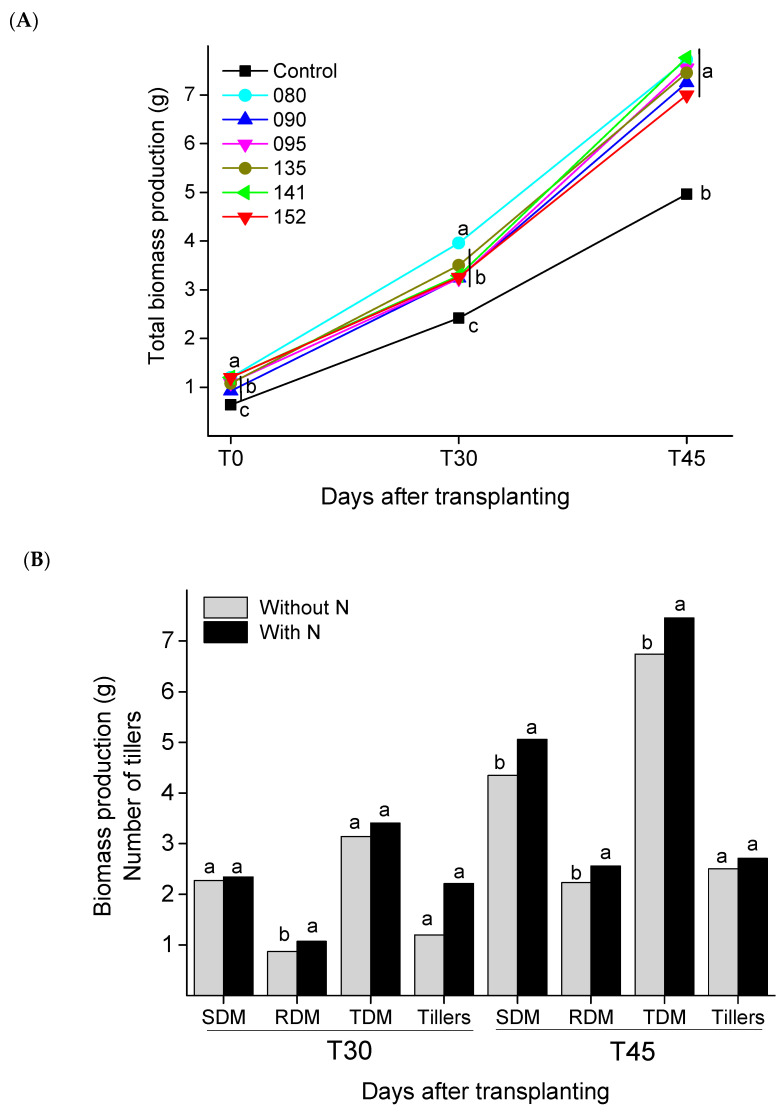
(**A**) Total biomass produced by sugarcane seedlings inoculated or not (Control) with six different bacterial strains (080, 090, 095, 135, 141, and 152) at day of transplanting (T0), 30 (T30) and 45 (T45). (**B**) Shoot dry matter (SDM), root dry matter (RDM), total dry matter (TDM), and number of tillers (Tillers) of sugarcane seedlings with (black bars) and without (gray bars) nitrogen fertilizer, 30 (T30) and 45 (T45) days after transplanting. Different letters (a–c) indicate significant differences among inoculation treatments within each analyzed time by the Scott-Knott test (*p* < 0.05).

**Figure 2 microorganisms-09-00479-f002:**
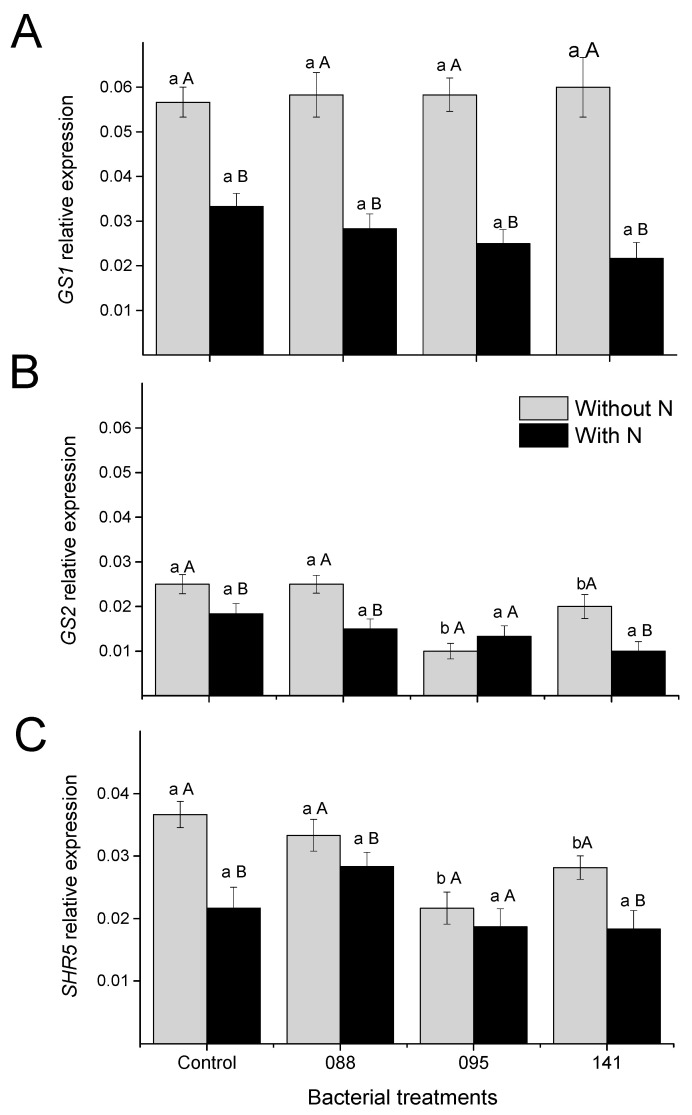
Relative expression levels of the genes *GS1* (**A**), *GS2* (**B**), and *SRH5* (**C**), in leaves of sugarcane seedlings inoculated or not (Control) with IAC-BECa-088, IAC-BECa-095, and IAC-BECa-141 = IAC/BECa-141 bacterial strains, and normalized by the expression of GAPDH reference gene transcript levels and with (black bars) or without (gray bars) nitrogen fertilizer, 45 days after transplanting. Different lower-case letters (a,b) indicate significant differences among bacterial treatments within each N fertilization treatment, and capital letters (A, B) indicate significant difference between N fertilization treatments within each bacterial treatment by the Scott-Knott test (*p* < 0.05).

**Figure 3 microorganisms-09-00479-f003:**
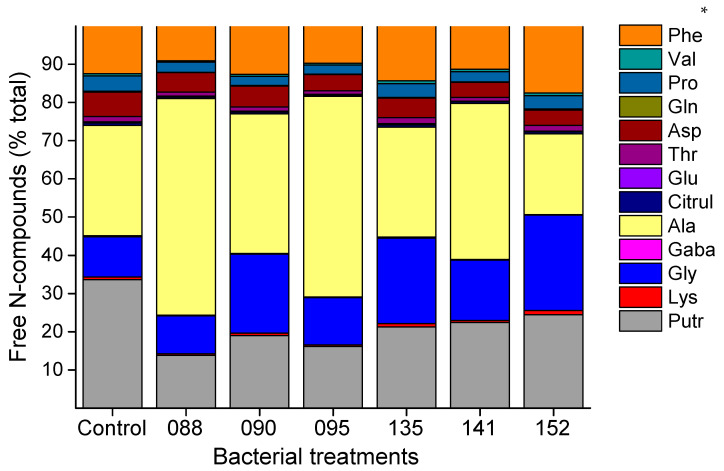
Main free amino acid and polyamine composition (% total) of leaves of sugarcane seedlings in function of the inoculation or not (Control) of six bacterial strains (IAC-BECa-088, IAC-BECa-090, IAC-BECa-095, IAC-BECa-135, IAC-BECa-141, IAC-BECa-152) 45 days after transplanting. * Phe, phenylalanine; Val, valine; Pro, Proline; Gln, glutamine; Asp, aspartate; Thr, threonine; Glu, glutamate; Citrul, citrulline; Ala, alanine; Gaba, gamma-aminobutyric acid; Gly, glycine; Lys, lysine; Putr, putrescine.

**Figure 4 microorganisms-09-00479-f004:**
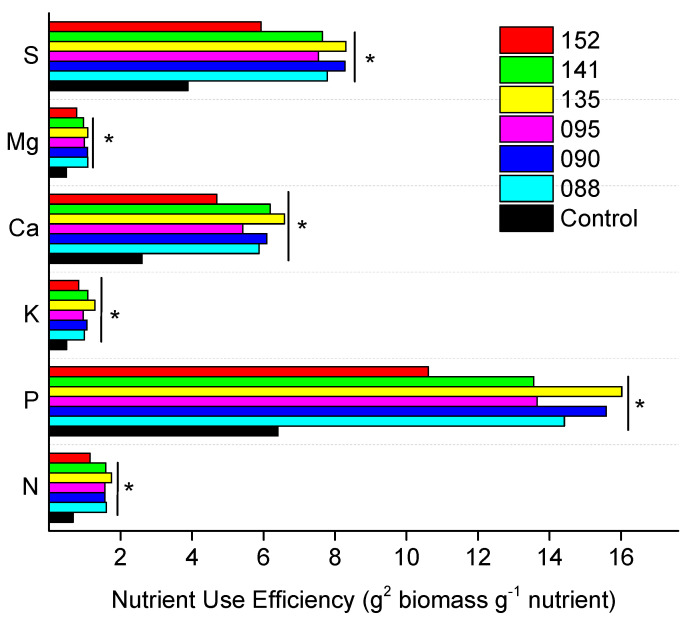
Nitrogen, phosphorus, potassium, calcium, magnesium, and sulphur use efficiency of sugarcane seedlings inoculated or not (Control) with six bacterial strains (IAC-BECa-088, IAC-BECa- 090, IAC-BECa-095, IAC-BECa-135, IAC-BECa-141, IAC-BECa-152) 45 days after transplanting. * Indicates significant differences among bacterial treatments by the Scott-Knott test (*p* < 0.05).

**Table 1 microorganisms-09-00479-t001:** Plant growth-promoting substances production (+) or not (−) by endophytic bacteria according literature.

Strain Code	Species Identity	*nifH*	IAA	PS	References
IAC/BECa-088	*Paraburkholderia caribensis*	+	−	−	[13,17]
IAC/BECa-090	*Kosakonia radicincitans*	+	−	−	[17]
IAC/BECa-095	*Kosakonia radicincitans*	+	+	−	[13]
IAC/BECa-135	*Paraburkholderia tropica*	−	−	−	[13,18]
IAC/BECa-141	*Pseudomonas fluorescens*	−	+	−	[1,13,18]
IAC/BECa-152	*Herbaspirillum frisingense*	−	+	−	[6,13,18,19]

Positive (+) or negative (−) signals mean positive or negative result for the *nifH* gene, indole acetic acid (IAA), and phosphate solubilization (PS).

**Table 2 microorganisms-09-00479-t002:** Shoot dry matter (SDM), root dry matter (RDM), total dry matter (TDM), and number of tillers of sugarcane seedlings inoculated or not (control) with six different bacterial inoculant day of transplanting (T0), 30 (T30) and 45 (T45) days after transplanting.

Time	T0							T30								T45							
Treatments	SDM		RDM		TDM		Tiller	SDM		RDM		TDM		Tiller		SDM		RDM		TDM		Tiller	
	g							g								g							
Control	0.27	c *	0.37	a	0.64	c	-	1.58	b	0.84	b	2.42	c	2.17	a	3.45	b	1.51	b	4.96	b	2.25	a
IAC/BECa-088	0.78	a	0.42	a	1.2	a	-	2.74	a	1.23	a	3.96	a	2.17	a	5.23	a	2.49	a	7.72	a	2.25	a
IAC/BECa-090	0.51	b	0.41	a	0.92	b	-	2.32	a	0.92	b	3.24	b	2	a	4.89	a	2.37	a	7.25	a	2.83	a
IAC/BECa-095	0.64	a	0.45	a	1.09	a	-	2.38	a	0.86	b	3.23	b	1.75	a	5.00	a	2.55	a	7.55	a	3.08	a
IAC/BECa-135	0.7	a	0.38	a	1.08	a	-	2.54	a	0.96	b	3.5	b	2.33	a	5.11	a	2.33	a	7.45	a	2.5	a
IAC/BECa-141	0.75	a	0.45	a	1.2	a	-	2.31	a	0.98	b	3.29	b	2.33	a	4.78	a	2.98	a	7.76	a	2.42	a
IAC/BECa-152	0.75	a	0.45	a	1.2	a	-	2.26	a	1.00	b	3.26	b	1.83	a	4.45	a	2.55	a	7	a	2.92	a
CV (%)	19.5		9.97		12.68		-	17.03		18.19		14.52		30.98		11.73		16.48		11.89		31.26	

* Different lower-case letters (a, b, c) indicate significant differences among inoculation treatments within each analyzed time by the Scott-Knott (*p* < 0.05). CV (%): coefficient of variation.

**Table 3 microorganisms-09-00479-t003:** Nutrient concentration, accumulation and nutrient-use efficiency index (UEI) of macro and micronutrients in plant shoot due to the bacterial treatments, 30 days after transplanting (T30) into the soil.

	Nutrient	N		P		K		Ca		Mg		S		Fe		Mn		Cu		Zn		B	
Nutrient Concentration	Treatments	g kg^−1^												mg kg^−1^									
	control	18.52	a *	1.83	b	27.38	b	4.73	a	2.72	a	2.85	a	68.80	c	57.80	a	8.50	c	32.08	b	1.58	b
	IAC-BECa-088	17.88	b	2.06	a	30.53	a	5.02	a	2.82	a	3.20	a	69.08	c	65.93	a	11.18	a	37.03	a	2.74	a
	IAC-BECa-090	19.28	a	1.68	b	27.03	b	5.03	a	2.90	a	2.92	a	80.43	b	66.70	a	9.15	b	33.58	a	2.33	a
	IAC-BECa-095	18.47	a	1.71	b	24.85	c	4.83	a	2.87	a	2.95	a	139.62	a	67.35	a	9.60	b	35.00	a	2.38	a
	IAC-BECa-135	17.62	b	1.78	b	25.03	c	4.78	a	2.80	a	2.97	a	79.24	b	59.51	a	9.00	b	28.99	b	2.54	a
	IAC-BECa-141	17.85	b	1.75	b	25.07	c	4.62	a	2.90	a	2.88	a	71.39	c	63.84	a	8.27	c	32.40	b	2.31	a
	IAC-BECa-152	18.19	b	1.43	c	21.80	d	4.12	b	2.65	a	2.55	a	69.78	c	56.72	a	7.90	c	30.18	b	2.26	a
	CV (%)	3.88		4.74		8.55		8.94		6.32		8.64		10.43		11.50		9.20		9.63		17.03	
Nutrient Accumulation	Treatments	g plant^−1^												mg plant^−1^									
	control	0.029	b	0.0029	d	0.0436	c	0.0075	b	0.0043	b	0.0045	b	0.109	c	0.0894	c	0.0135	c	0.0508	c	0.0193	a
	IAC-BECa-088	0.049	a	0.0056	a	0.0836	a	0.0137	a	0.0077	a	0.0088	a	0.189	b	0.1803	a	0.0308	a	0.1012	a	0.0173	a
	IAC-BECa-090	0.045	a	0.0039	c	0.0626	b	0.0116	a	0.0067	a	0.0068	a	0.186	b	0.1548	a	0.0212	b	0.0779	b	0.0190	a
	IAC-BECa-095	0.044	a	0.0041	c	0.0591	b	0.0115	a	0.0068	a	0.0070	a	0.328	a	0.1589	a	0.0228	b	0.0833	b	0.0202	a
	IAC-BECa-135	0.045	a	0.0046	b	0.0645	b	0.0122	a	0.0071	a	0.0076	a	0.205	b	0.1522	a	0.0230	b	0.0745	b	0.0200	a
	IAC-BECa-141	0.041	a	0.0040	b	0.0582	b	0.0107	a	0.0067	a	0.0067	a	0.164	b	0.1480	a	0.0191	b	0.0755	b	0.0191	a
	IAC-BECa-152	0.041	a	0.0032	d	0.0494	c	0.0093	b	0.0059	a	0.0058	b	0.156	b	0.1266	b	0.0177	c	0.0682	c	0.0218	a
	CV (%)	15.67		17.12		20.96		18.30		16.07		20.43		15.79		18.21		20.34		20.99		19.24	
UEI	Treatments	g^2^ biomass g^−1^ nutrient												mg^2^ biomass g^−1^ nutrient									
	control	0.1480	b	1.4716	b	0.0973	b	0.5755	b	0.9900	b	0.9783	b	0.0384	b	0.0489	b	0.3127	b	0.0833	b	0.0525	a
	IAC-BECa-088	0.4211	a	3.6949	a	0.2476	a	1.5051	a	2.6804	a	2.3511	a	0.1096	a	0.1143	a	0.6745	a	0.2037	a	0.1309	a
	IAC-BECa-090	0.2798	a	3.2127	a	0.2025	a	1.0953	a	1.8673	a	1.8493	a	0.0676	a	0.0810	a	0.5901	a	0.1608	a	0.1038	a
	IAC-BECa-095	0.3087	a	3.3297	a	0.2283	a	1.1819	a	1.9828	a	1.9213	a	0.0450	b	0.0874	a	0.6005	a	0.1622	a	0.1171	a
	IAC-BECa-135	0.3763	a	3.7250	a	0.2620	a	1.3904	a	2.4004	a	2.2113	a	0.0839	a	0.1112	a	0.7395	a	0.2283	a	0.1358	a
	IAC-BECa-141	0.3084	a	3.1453	a	0.2190	a	1.1985	a	1.8927	a	1.9120	a	0.0780	a	0.0856	a	0.6677	a	0.1677	a	0.1039	a
	IAC-BECa-152	0.2926	a	3.7625	a	0.2413	a	1.3073	a	2.0530	a	2.0769	a	0.0786	a	0.0977	a	0.6852	a	0.1779	a	0.1238	a
	CV (%)	34.51		35.53		31.80		36.22		36.64		32.52		37.84		38.72		36.52		31.19		46.70	

* Different lower-case letters (a, b, c, d) indicate significant differences among inoculation treatments by Scott-Knott (*p* < 0.05). CV (%): coefficient of variation.

**Table 4 microorganisms-09-00479-t004:** Nutrient concentration, accumulation and nutrient-use efficiency index (UEI) of macro and micronutrients in plant shoot due to the bacterial treatments, 45 days after transplanting (T45) into the soil.

	Nutrient	N		P		K		Ca		Mg		S		Fe		Mn		Cu		Zn		B	
Nutrient Concentration	Treatments	g kg^−1^												mg kg^−1^									
	control	18.22	a *	1.93	b	25.05	c	4.70	a	2.52	a	3.29	c	60.11	b	43.02	b	7.38	b	25.26	b	17.16	c
	IAC-BECa-088	18.60	a	1.95	b	30.28	a	4.78	a	5.57	a	3.76	a	60.28	b	50.60	a	8.57	a	28.01	a	23.93	a
	IAC-BECa-090	16.49	b	1.68	c	24.03	c	4.07	b	2.28	a	3.08	c	63.53	a	43.71	b	6.88	b	21.92	b	19.81	b
	IAC-BECa-095	16.79	b	1.91	b	27.28	b	4.83	a	2.55	a	3.51	b	60.90	b	52.46	a	7.23	b	27.63	a	24.06	a
	IAC-BECa-135	16.28	b	1.85	b	22.12	d	4.53	a	2.55	a	3.48	b	59.84	b	52.83	a	8.03	a	29.68	a	16.95	c
	IAC-BECa-141	16.08	b	1.72	c	21.78	d	3.92	b	2.48	a	3.05	c	52.02	b	4.16	b	6.91	b	22.70	c	20.25	b
	IAC-BECa-152	19.03	a	2.18	a	24.45	c	4.23	b	2.58	a	3.80	a	71.20	a	53.70	a	8.73	a	22.99	c	21.48	a
	CV (%)	4.61		5.87		7.10		6.16		8.54		7.34		11.67		5.46		7.57		7.75		11.75	
Nutrient Accumulation	Treatments	g plant^−1^												mg kg^−1^									
	control	0.0633	c	0.5206	b	0.0863	d	0.0162	c	0.0087	c	0.0114	c	0.2254	b	0.1483	c	0.0256	d	0.0847	c	0.0593	c
	IAC-BECa-088	0.0971	a	0.5148	b	0.1582	a	0.0249	a	0.0134	a	0.0197	a	0.3161	a	0.2641	a	0.0448	a	0.1462	a	0.1258	a
	IAC-BECa-090	0.0796	b	0.6001	a	0.1162	c	0.0198	b	0.0111	b	0.1503	b	0.3199	a	0.2141	b	0.0328	c	0.1064	b	0.0972	b
	IAC-BECa-095	0.0845	b	0.5248	b	0.1377	b	0.0241	a	0.0128	a	0.0175	a	0.3055	a	0.2612	a	0.0360	b	0.1360	a	0.1213	a
	IAC-BECa-135	0.0827	b	0.5419	b	0.1127	c	0.0231	a	0.0129	a	0.0178	a	0.3046	a	0.2693	a	0.0409	a	0.1505	a	0.0853	b
	IAC-BECa-141	0.0772	b	0.5835	b	0.1040	c	0.0185	b	0.0118	b	0.0147	b	0.2494	b	0.2203	b	0.3325	c	0.1094	b	0.0955	b
	IAC-BECa-152	0.0845	b	0.4610	c	0.1088	c	0.0189	b	0.0115	b	0.0170	a	0.3154	a	0.2391	b	0.3832	b	0.1021	b	0.0938	b
	CV (%)	11.56		8.63		13.20		10.32		11.51		14.38		18.34		12.39		12.29		11.93		16.16	
UEI	Treatments	g^2^ biomass g^−1^ nutrient												mg^2^ biomass g^−1^ nutrient									
	control	0.6750	b	6.4070	b	0.4880	b	2.6050	b	0.4856	b	3.8820	b	0.1720	c	0.2680	b	1.5730	b	0.4690	b	0.6930	b
	IAC-BECa-088	1.6050	a	14.4150	a	0.9880	a	5.8750	a	1.0817	a	7.7800	a	0.4620	a	0.5680	a	3.5540	a	1.0850	a	1.1920	b
	IAC-BECa-090	1.5650	a	15.5860	a	1.0640	a	6.0940	a	1.0778	a	8.2790	a	0.4080	a	0.5790	a	3.9640	a	1.2080	a	1.3930	b
	IAC-BECa-095	1.5650	a	13.6540	a	0.9560	a	5.4150	a	0.9898	a	7.5390	a	0.4170	a	0.5100	a	3.7390	a	1.0180	a	1.0910	b
	IAC-BECa-135	1.7420	a	16.0180	a	1.2870	a	6.5820	a	1.0806	a	8.3050	a	0.4890	a	0.5680	a	3.8180	a	1.1350	a	1.8800	b
	IAC-BECa-141	1.5860	a	13.5540	a	1.0810	a	6.1860	a	0.9669	a	7.6490	a	0.4510	a	0.5210	a	3.5250	a	1.1150	a	1.3350	a
	IAC-BECa-152	1.1430	b	10.6090	b	8.3200	a	4.6940	a	0.7745	b	5.9290	b	0.3140	b	0.4060	b	2.1560	b	0.8450	a	1.0880	b
	CV (%)	32.01		30.79		27.89		31.73		29.27		27.69		28.30		28.21		35.49		39.11		34.95	

* Different lower-case letters (a, b, c, d) indicate significant differences among inoculation treatments by Scott-Knott (*p* < 0.05). CV (%): coefficient of variation.

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
