# Peer review of "Plant-Growth Endophytic Bacteria Improve Nutrient Use Efficiency and Modulate Foliar N-Metabolites in Sugarcane Seedling"

_microorganisms, 2021, doi:10.3390/microorganisms9030479_

Round 1

Reviewer 1 Report

The  manuscript entitled "Plant-growth endophytic bacteria improve nutrient use efficiency and modulate foliar N-metabolites in sugarcane seedling" deal  nutrient acquisition efficiency an important attributes of endophytic  microorganisms. The present manuscript has merit  for publication in this journal.

Author Response

Dear reviewer, 

thank you for indicating the manuscript for publication.

Best regards,

Cipriano et al.

Reviewer 2 Report

The manuscript “Plant-growth endophytic bacteria improve nutrient use 2 efficiency and modulate foliar N-metabolites in sug-3 arcane seedling” it’s a research study was to evaluate the effect of bacterial inoculants on the development of sugarcane plants, and the physiological and biochemical aspects of plant-bacteria interaction.

Six strains used.

It’s a very easy to read for the reader with good language and presentation.

A few comments:

  1. At line 161.  The soil chemical analysis is described in Table 2 .. BUT Table 2 present plant properties. Pls correct
  2. At Table 2 (line 180) there are not units for the properties (maybe gram But pls add units in Tables.
  3. At line 190. .. nutrient content and nutrient-use efficiency index (UEI) were estimated 190 according to Siddiq and Glass (23).

According  “Siddiq and Glass 1981“ used different abbreviations   (E and I in papers ) . Authors must to give at the same abbreviation because confuse the reader. For example at table 4 the authors use the term “Efficiency Index of use” BUT at line 190 nutrient-use efficiency index (UEI).

  1. At line 253 & 255 the site link must to add as references at refs.
  2. I suggest to authors to give more details how to estimate the “Efficiency of Utilization” and especially the “Utilization Index”

How estimated the index? The index its comparison , for example,  between two varieties (  Siddiq and Glass 1981“) or two different treatments ?

I=E1/E2= W2/C2   W1/C1= W2/W1  X C1/C2    (Siddiq and Glass 1981)

Where w2, or w1 c1 and c2? Please explain.

  1. There are not abbreviation at the end of manuscript (pls add)
